# Nucleotide biosynthesis links glutathione metabolism to ferroptosis sensitivity

Amy Tarangelo[1] , Jason Rodencal[1] , Joon Tae Kim[2] , Leslie Magtanong[1] , Jonathan Z Long[2], Scott J Dixon[1]

**Nucleotide synthesis is a metabolically demanding process essential for DNA replication and other processes in the cell. Several anti-cancer drugs that inhibit nucleotide metabolism induce apoptosis. How inhibition of nucleotide metabolism impacts non-apoptotic cell death is less clear. Here, we report that inhibition of nucleotide metabolism by the p53 pathway is sufficient to suppress the non-apoptotic cell death process of ferroptosis. Mechanistically, stabilization of wild-type p53 and induction of the p53 target gene *CDKN1A* (p21) leads to decreased expression of the ribonucleotide reductase (RNR) subunits *RRM1* and *RRM2*. RNR is the rate-limiting enzyme of de novo nucleotide synthesis that reduces ribonucleotides to deoxyribonucleotides in a glutathione-dependent manner. Direct inhibition of RNR results in conservation of intracellular glutathione, limiting the accumulation of toxic lipid peroxides and preventing the onset of ferroptosis in response to cystine deprivation. These results support a mechanism linking p53-dependent regulation of nucleotide metabolism to non-apoptotic cell death.**

## Introduction

Nucleotides are required for various metabolic processes in the cell, including energy metabolism, phospholipid synthesis, N-glycosylation, and of course the synthesis of DNA and RNA (Lane & Fan, 2015). De novo nucleotide synthesis in mammalian cells is accomplished through a multi-step pathway that ultimately converts ribonucleotide triphosphates to deoxyribonucleotide triphosphates (dNTPs) (Elledge et al, 1992; Tran et al, 2019). dNTPs are essential for DNA replication. The conversion of NTPs to dNTPs is catalyzed by ribonucleotide reductase (RNR), a heterodimer of RRM1 and RRM2 or RRM2B subunits. The reaction catalyzed by RNR requires reducing equivalents supplied by (reduced) glutathione (GSH) or thioredoxin to complete the catalytic cycle, with the GSH/glutaredoxin system being most important for mammalian RNR reduction and function (Zahedi Avval & Holmgren, 2009; Sengupta et al, 2019). RNR subunit expression is controlled by the cell cycle machinery to match the need for

nucleotide synthesis to cell proliferation (Eriksson et al, 1984; Sigoillot et al, 2003). Thus, cell cycle progression requires redox-regulated RNR function. Whether disruption of cell cycle progression alters other redox-dependent processes in the cell is poorly understood.

The p53 tumor suppressor pathway is an important regulator of cell cycle progression and nucleotide metabolism (Sherley, 1991; Zhang et al, 2001; Huang et al, 2018). Activation of the p53 pathway may limit oxidative stress and promote cell survival during serine deprivation, possibly by modulating nucleotide synthesis (Maddocks et al, 2013). However, the specific mechanism involved in this protective effect is unclear. More generally, how p53 pathway-dependent regulation of nucleotide metabolism links to the broader control of cell fate by p53 requires more detailed investigation.

In addition to its well-established role in apoptosis, the p53 pathway regulates ferroptosis sensitivity (Jiang et al, 2015; Kruiswijk et al, 2015; Xie et al, 2017; Tarangelo et al, 2018; Chu et al, 2019; Venkatesh et al, 2020a). Ferroptosis is an oxidative, iron-dependent form of cell death important for tumor suppression and pathological cell death in mammals (Linkermann et al, 2014; Skouta et al, 2014; Fang et al, 2019; Badgley et al, 2020; Ubellacker et al, 2020). Ferroptosis is characterized by the toxic accumulation of lipid peroxides at the cell membrane (Dixon et al, 2012; Stockwell et al, 2017). Lipid peroxide accumulation is inhibited by glutathione peroxidase 4 (GPX4), a lipid hydroperoxidase that uses reduced GSH as a cofactor (Friedmann Angeli et al, 2014; Yang et al, 2014). Ferroptosis can therefore be induced by depriving cells of the sulfur-containing GSH precursor, cysteine, via inhibition of the system $x_c^-$ cystine/glutamate antiporter (e.g., using small molecules of the erastin family), or by direct inhibition of GPX4 itself (Forcina & Dixon, 2019). Stabilization of wild-type p53 and induction of p21 can decrease the susceptibility of cancer cells to ferroptosis by conserving intracellular GSH (Tarangelo et al, 2018). Overexpression of p21 alone may also be sufficient to inhibit ferroptosis via interaction with cyclin dependent kinases (CDKs) in some contexts (Venkatesh et al, 2020b). However, the specific mechanism by which p53 and p21 conserve GSH to inhibit ferroptosis is not clear.

Here, we explored how stabilization of p53 and induction of p21 promote ferroptosis resistance in human cancer cells. Using

[1]Department of Biology, Stanford University, Stanford, CA, USA  [2]Department of Pathology and Stanford, Chemistry, Engineering and Medicine for Human Health (ChEM-H), Stanford University School of Medicine, Stanford, CA, USA

Correspondence: sjdixon@stanford.edu
Amy Tarangelo's present address is Children's Medical Center Research Institute, UT Southwestern, Dallas, TX, USA.

 

transcriptomic analysis, we find that activation of the p53–p21 pathway down-regulates expression of many genes involved in nucleotide metabolism. Among the most down-regulated genes in this subset are the RNR subunits *RRM1* and *RRM2*. Using cell death kinetic analysis, gene-editing and metabolic tracing, we show that inhibition of RNR-dependent nucleotide metabolism allows for conservation of intracellular GSH and a reduction in ferroptosis susceptibility in cancer cells. Thus, regulation of nucleotide synthesis by the p53–p21 axis provides a crucial link between GSH metabolism and ferroptosis susceptibility.

## Results

### Pharmacological p53 stabilization inhibits ferroptosis

The effects of p53 on ferroptosis are controversial, with evidence that p53 may induce or suppress this cell death process in a context-dependent manner (Jiang et al, 2015; Xie et al, 2017; Tarangelo et al, 2018). We examined the effects of p53 manipulation in human HT-1080 fibrosarcoma cells, which express wild-type p53. We inhibited the function of the p53 negative regulator human double minute 2 (HDM2) using the potent and specific small molecule inhibitors nutlin-3 and MI-773/SAR405838 (Vassilev et al, 2004; Wang et al, 2014). In all experiments cell death was quantified using the scalable time-lapse analysis of cell death kinetics (STACK) imaging method (Forcina et al, 2017). Using STACK, we determined the precise time of cell death onset within each population, denoted as $D_O$. Consistent with previous results (Tarangelo et al, 2018), nutlin-3 and MI-773 stabilized p53 levels and delayed the onset of ferroptosis in response to erastin2 treatment, whereas an inactive enantiomer of nutlin-3 (nutlin-3b) had no such effects (Fig 1A–D). Cell death in these experiments was suppressed by co-treatment with the specific ferroptosis inhibitor ferrostatin-1, confirming that p53 stabilization did not alter the mode of cell death induced by erastin2 (Fig 1C). The effects of nutlin-3 were dose-dependent, with increasing concentrations of this inhibitor resulting in greater p53 stabilization, which correlated with increased induction of the p53 target genes *CDKN1A* and *HDM2* and greater suppression of erastin2-induced ferroptosis (Fig S1A–D). Collectively, these results show that pharmacological HDM2 inhibition robustly delays the onset of ferroptosis in HT-1080 cells through an on-target mechanism.

### p21 is necessary but not sufficient to suppress ferroptosis

We previously showed that expression of *CDKN1A*, encoding the CDK inhibitor p21, was necessary for wild-type p53 to suppress ferroptosis (Tarangelo et al, 2018). Recently it was suggested that p21 expression alone may be sufficient to suppress ferroptosis in some contexts, independent of p53 (Venkatesh et al, 2020b). To examine whether p21 expression was sufficient to fully recapitulate the functions of p53 in ferroptosis suppression in HT-1080 cells, we generated cell lines expressing Flag-tagged *CDKN1A* under the control of a doxycycline (Dox)-inducible promoter, denoted p21[Dox]. We introduced p21[Dox] into both HT-1080 Control (i.e., wild-type) and p53 gene disrupted (KO) (p53[KO]) backgrounds (Tarangelo et al, 2018). Dox treatment induced robust expression of full-length (FL) p21 in both Control and p53[KO] cells, which was not further augmented by co-treatment with nutlin-3 (Fig 2A). Dox pretreatment alone delayed the onset of ferroptosis (i.e., increased $D_O$) in response to erastin2 in Control cells capable of expressing p21, albeit not to the same extent as nutlin-3 treatment (Fig 2B and C). By contrast, in p53[KO] cells, Dox-induced p21 induction had little ability to prevent ferroptosis (Fig 2B and C). Thus, p21 expression alone is not sufficient to fully recapitulate the effects of p53 stabilization, especially when cells lack wild-type p53.

We further dissected the contribution of different p21 functional domains to ferroptosis suppression. In particular, we focused on the p21 N-terminal cyclin-CDK binding domain, required for inhibition of G1/S cell cycle progression, and the C-terminal PCNA binding domain involved in suppressing DNA synthesis (Chen et al, 1995)

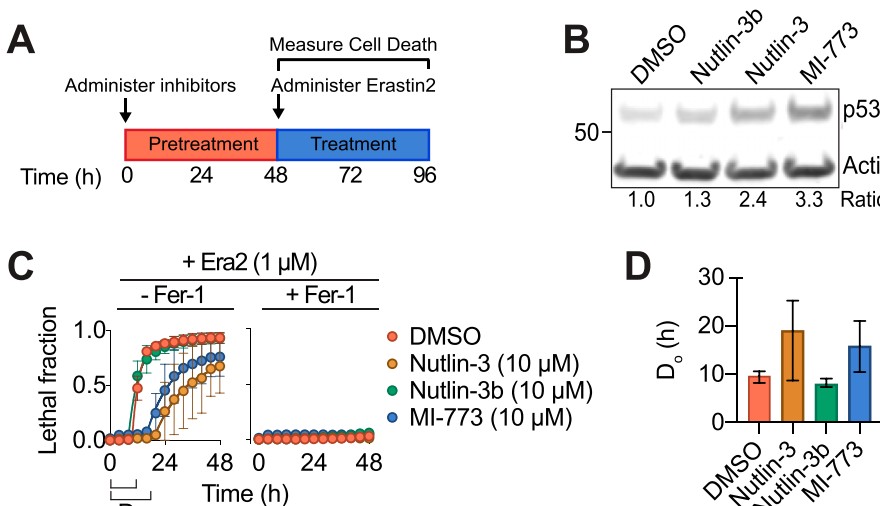

**Figure 1. Diverse p53-activating stimuli suppress ferroptosis.**
**(A)** Schematic of experimental procedures for assaying cell death susceptibility. **(B)** Western blot for HT-1080 Control cells treated with HDM2 inhibitors (10 $\mu$M, 48 h). The ratio refers to p53 levels, normalized to the Actin loading control. **(C)** Lethal response curves for HT-1080 Control cells expressing a nuclear mKate2 signal (HT-1080[N]) pretreated with HDM2 inhibitors (10 $\mu$M, 48 h) followed by treatment with Erastin2 (Era2, 1 $\mu$M) ± the specific ferroptosis inhibitor ferrostatin-1 (Fer-1, 1 $\mu$M). **(C, D)** Quantification of the time of death onset ($D_O$) for the lethality curves shown in (C), calculated using scalable time-lapse analysis of cell death kinetics. Data information: all data are plotted as the mean ± SD at least three independent experiments, except for $D_O$ values, which are mean ± 95% confidence interval.

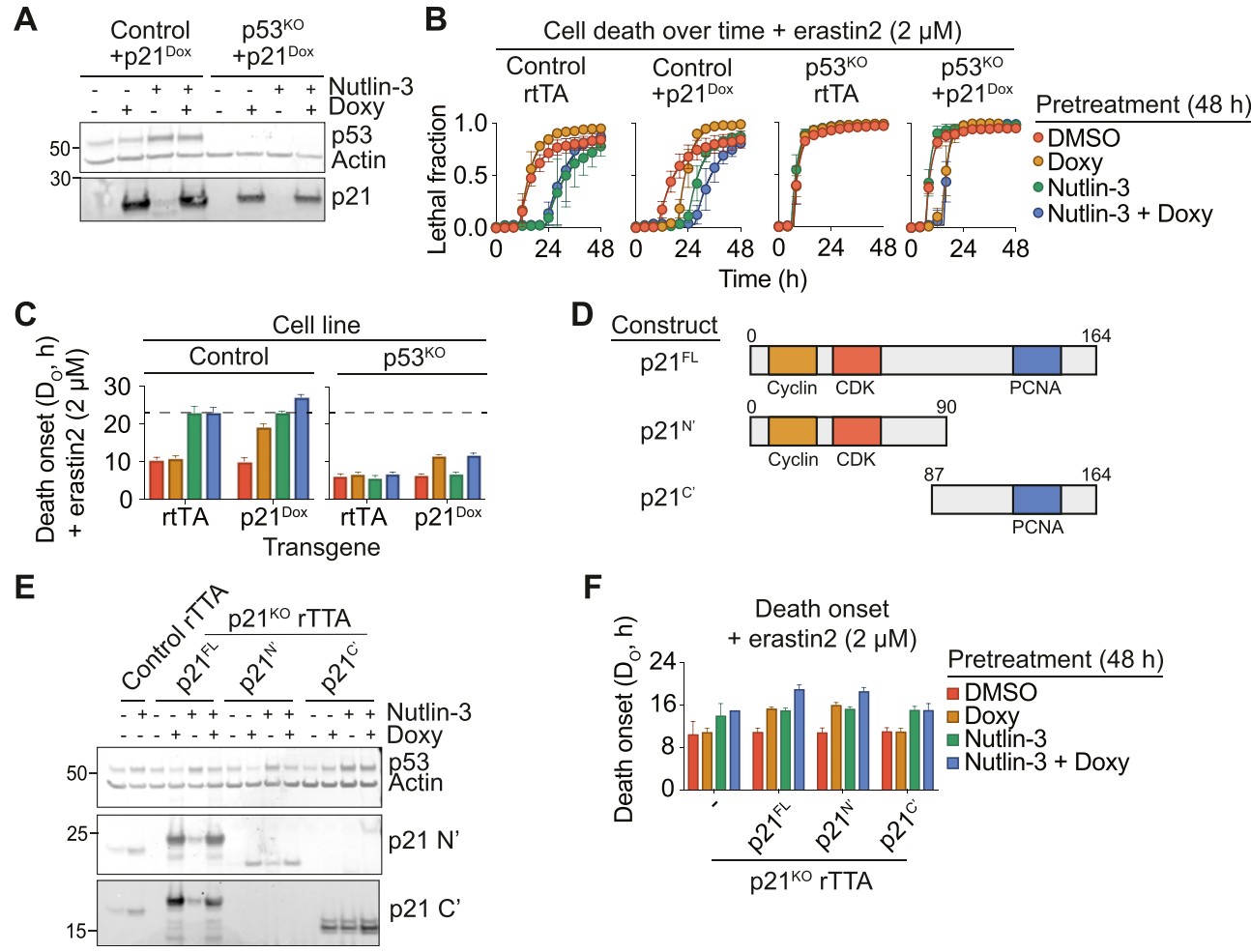

**Figure 2.   The cyclin-dependent kinase-binding domain of p21 is required to suppress ferroptosis.**
**(A)** Protein expression in HT-1080 Control or p53[KO] + p21[Dox] treated ± nutlin-3 (10 $\mu$M) ± doxycycline (Doxy, 1 $\mu$g/ml) for 48 h. **(B)** Cell death in HT-1080[N] Control or p53[KO] cells ± the p21[Dox] cassette, pretreated ± nutlin-3 (10 $\mu$M) ± Doxy (1 $\mu$g/ml) for 48 h, followed by treatment with Erastin2 (2 $\mu$M). **(B, C)** Timing of death onset ($D_O$) calculated from data shown in (B). **(D)** Schematic of the structure of p21 full length (FL), N terminal fragment (N'), or C terminal fragment (C'). **(E)** Protein expression in HT-1080 stable cell lines ± nutlin-3 (10 $\mu$M) ± Doxy (1 $\mu$g/ml) for 48 h. **(F)** Cell death onset ($D_O$) plotted for the indicated conditions. Data information: In (B), data represent mean ± SD from three independent experiments. In (C, F), data represent mean ± 95% confidence interval from three independent experiments.

(Fig 2D). We introduced Dox-inducible FL Flag-tagged p21, or isolated N and C terminal truncation mutants (p21[N'] and p21[C'], respectively), into HT-1080 p21 KO cells (Tarangelo et al, 2018). Dox induction of all constructs was validated with antibodies specific for epitopes in the p21 N and C termini (Fig 2E). Consistent with negative feedback by p21 on p53 levels (Broude et al, 2007), induction of p21[FL] and p21[N'], but not p21[C'], decreased basal p53 levels and attenuated p53 stabilization upon nutlin-3 treatment, further validating the function of these proteins (Fig 2E). With respect to cell death, induction of p21[FL] modestly delayed the onset of erastin2-induced death alone, and to a greater extent when combined with nutlin-3, consistent with an independent requirement for both p53 and p21 in the suppression of ferroptosis (Fig 2F). Induction of p21[N'], but not p21[C'], likewise modestly suppressed ferroptosis alone and more potently suppressed ferroptosis together with nutlin-3 treatment, to the same extent as p21[FL]. Thus, the p21 N terminus is required for p21 to suppress ferroptosis.

## p53 does not modulate ferroptosis via GSH export or altered electron transport chain function

Activation of the p53–p21 pathway can suppress ferroptosis by conserving intracellular GSH (Tarangelo et al, 2018). The specific mechanism accounting for GSH conservation is unclear. To investigate further, we initially pursued a hypothesis-driven approach. First, we investigated the role of *ABCC1*, which encodes the multidrug-resistance protein 1 (MRP1). MRP1 exports GSH from the cell and loss of MRP1 expression can thereby conserve intracellular GSH and delay the onset of ferroptosis, much like p53 stabilization (Cole, 2014; Cao et al, 2019). Accordingly, we investigated whether p53 stabilization delayed ferroptosis and conserved intracellular GSH by down-regulating *ABCC1* expression. However, nutlin-3 treatment did not decrease *ABCC1* expression in HT-1080 Control cells or in p53 wild-type U-2 OS osteosarcoma cells (Fig S2A). Indeed, whereas nutlin-3 pretreatment suppressed erastin2-induced

cell death in U-2 OS cells expressing an empty control vector, this condition enhanced ferroptosis in U-2 OS cells overexpressing MRP1 (Fig S2B). Thus, p53 did not appear to suppress ferroptosis by reducing *ABBC1*/MRP1 function.

Mitochondria are a major source of intracellular reactive oxygen species (ROS), whose accumulation is buffered by GSH-dependent enzymes (Handy et al, 2009; Marí et al, 2009). Mitochondrial oxidative phosphorylation has been suggested to promote ferroptosis through the production of lipid peroxides (Gao et al, 2019). p53 activation can suppress the mitochondrial tricarboxylic acid cycle and oxidative phosphorylation through inhibition of malic enzymes 1/2 or by inducing a PUMA-dependent metabolic switch (Jiang et al, 2013; Kim et al, 2019). Accordingly, we hypothesized that p53 stabilization conserved GSH and suppressed ferroptosis by reducing mitochondrial activity. To investigate this, we generated HT-1080 cells lacking mitochondrial DNA (i.e., $\rho^0$ cells) via long-term incubation in ethidium bromide. Compared with control (i.e., $\rho^+$) cells, $\rho^0$ cells expressed mitochondrial genes at undetectable levels and were entirely deficient in oxidative metabolism, as detected using Seahorse technology (Fig S2C and D). However, there was no difference in basal sensitivity to erastin2–induced ferroptosis in $\rho^0$ compared to $\rho^+$ cells, and pretreatment with nutlin-3 suppressed ferroptosis equally well in both lines (Fig S2E). These results indicated that the ability of the p53 pathway to suppress ferroptosis was unlikely to involve changes in mitochondrial oxidative phosphorylation.

## The p53–p21 axis regulates nucleotide metabolic gene networks

We next took an unbiased approach to identify candidate pathways impacting GSH metabolism and ferroptosis sensitivity downstream of p53 and p21. Towards this end, we performed RNA sequencing of HT-1080 Control, p53$^{KO}$ and p21$^{KO}$ cells treated with or without nutlin-3 (10 $\mu$M, 48 h). In Control cells, nutlin-3 treatment resulted in down-regulation of 2,714 genes and up-regulation of 2,429 genes (Fig 3A). Consistent with expectations, bona fide p53 targets including *HDM2* and *CDKN1A* were among the most highly up-regulated genes in Control cells, whereas genes known to be negatively regulated by p53, such as *E2F2*, were strongly down-regulated (el-Deiry et al, 1993; Chen et al, 1994; Timmers et al, 2007). Compared with Control cells, p53$^{KO}$ cells exhibited far fewer nutlin-3–induced changes in gene expression, consistent with most changes observed in Control cells being p53-dependent (Fig 3A). By contrast, in p21$^{KO}$ cells, nutlin-3 treatment resulted in the differential expression of even more genes than observed in Control cells, consistent with the notion that p21 is itself an important transcriptional regulator (Fig 3A) (Delavaine & La Thangue, 1999; Ferrándiz et al, 2012). We exploited this large RNA Seq dataset to develop specific hypotheses concerning the regulation of ferroptosis by the p53 pathway.

In Control cells, nutlin-3 treatment down-regulated numerous genes involved in nucleotide metabolism, including *DTYMK*, *CTPS1*, *RRM1*, and *RRM2* (Fig 3B). *RRM1* and *RRM2* encode the R1 and R2 subunits of the heterodimeric enzyme RNR. RNR catalyzes the reduction of ribonucleotides to deoxyribonucleotides. In mammalian cells, RNR activity requires GSH (Zahedi Avval & Holmgren, 2009; Sengupta et al, 2019). Activation of the p53/p21 pathway can enable cancer cells to resist oxidative stress, and this effect may

involve inhibition of nucleotide synthesis (Maddocks et al, 2013). We therefore focused on the regulation of RNR function by the p53 pathway. Using RT-qPCR we confirmed that both *RRM1* and *RRM2* were down-regulated after nutlin-3 treatment, with *RRM2* down-regulation being entirely p53-dependent and *RRM1* regulated by both p53 and p21 (Fig 3C). At the protein level, R2 expression was more sensitive than expression of R1 to down-regulation upon p53 stabilization (Fig 3D). Consistent with decreased RNR activity, treatment of Control cells with nutlin-3 or MI-773 decreased levels of dAMP, a metabolite downstream of RNR (Fig 3E). Thus, activation of the p53–p21 pathway broadly down-regulates nucleotide metabolism, including RNR expression and activity.

## RNR inhibition blocks ferroptosis

RNR is oxidized as part of its catalytic mechanism. In mammalian cells, reduction of oxidized RNR requires GSH (Sengupta et al, 2019). Accordingly, we hypothesized that inhibition of RNR function may suppress ferroptosis by reducing the draw on the existing GSH pool, which can then be re-routed to suppress ferroptosis. To test this hypothesis, we first treated cells with small molecule inhibitors of pyrimidine and purine metabolism, which generate the ribonucleotide precursors that RNR reduces to dNTPs in a GSH-dependent manner. Cells were pretreated with mycophenolic acid (MPA), to inhibit the purine synthetic enzyme inosine-5'-monophosphate dehydrogenase, or with pyrazofurin (Pyr), to inhibit the pyrimidine synthetic enzyme orotidine 5'-monophosphate decarboxylase (UMPS) (Ringer et al, 1991; Fleming et al, 1996). Nucleotide depletion itself triggers p53 stabilization, and we confirmed that both MPA and Pyr caused this effect in HT-1080 Control cells (Fig 4A). Consistent with our hypothesis, 24 h pretreatment with either MPA or Pyr protected Control cells from erastin2-induced cell death (Fig 4B). Moreover, unlike with nutlin-3 (Fig 1C) (Tarangelo et al, 2018), MPA and Pyr pretreatment protected both p53$^{KO}$ and p21$^{KO}$ cells from erastin2-induced cell death, consistent with RNR acting downstream of the p53/p21 axis. Crucially, in all three cell lines, the de novo GSH synthesis inhibitor buthionine sulfoximine (BSO) abolished the ability of MPA and Pyr pretreatment to inhibit ferroptosis, consistent with a GSH-dependent protective mechanism (Fig 4B). Of note, the restoration of cell death by BSO co-treatment also indicates that MPA and Pyr were not inhibiting ferroptosis in an off-target manner by acting as radical trapping antioxidants (Conlon et al, 2021).

RNR function can be directly inhibited by the small molecules gemcitabine (Gem) and hydroxyurea (HU), which serve as analogs of cytidine to inhibit dNTP synthesis (Krakoff et al, 1968; Heinemann et al, 1990). Gem and HU did not strongly activate p53 and/or induce p21 expression in HT-1080 Control, p53$^{KO}$ or p21$^{KO}$ cells (Fig 4C). However, pretreatment with Gem or HU strongly suppressed ferroptosis in all three cell lines (Fig 4D). Pretreatment with Gem, but not nutlin-3, also delayed the onset of ferroptosis in p53 null H1299 cell lines (Fig S3A and B).

The effect of Gem pretreatment on ferroptosis in HT-1080 cells was reversed by co-treatment with BSO, consistent with a GSH-dependent mechanism (Fig 4E). Further consistent with this mechanism, Gem pretreatment did not suppress ferroptosis induced by the direct GPX4 inhibitor ML162, whose ability to induce

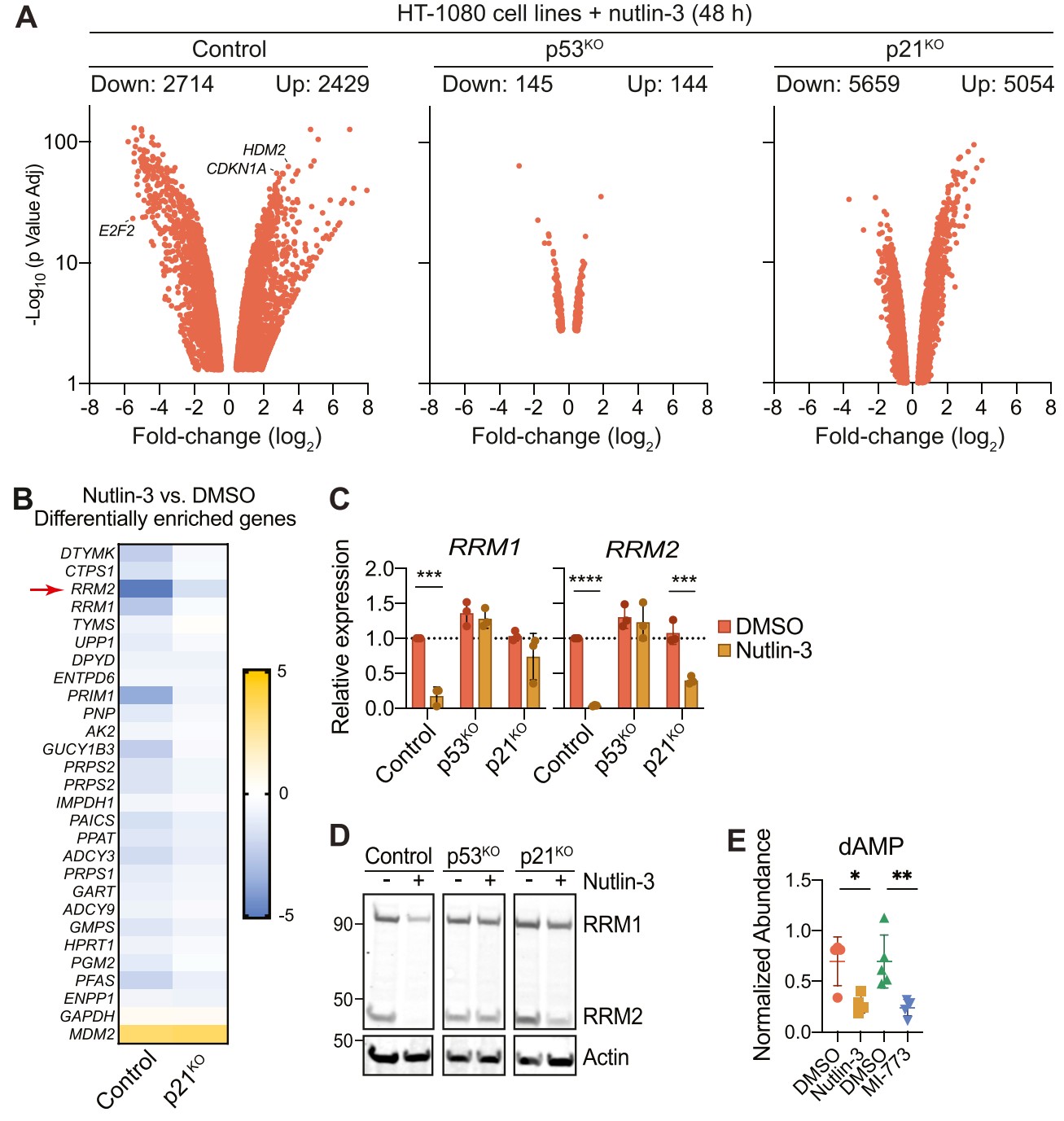

**Figure 3. p53 stabilization reduces ribonucleotide reductase expression and activity.**
**(A)** Volcano plot summarizing significantly ($P < 0.05$) up-regulated and down-regulated genes in HT-1080 Control, p53[KO] and p21[KO] cells treated ± nutlin-3 (10 μM, 48 h). RNA sequencing data are the average of two independent experiments. **(B)** Differential gene expression summary for genes with annotated functions in nucleotide metabolism from cells treated and analyses as described in (A). Legend scale is $\log_2$ fold-change (nutlin-3/DMSO). **(C)** mRNA levels in HT-1080 Control, p53[KO], and p21[KO] cells treated ± nutlin-3 (10 μM, 48 h) determined using RT-qPCR. **(D)** Protein expression in HT-1080 Control, p53[KO], and p21[KO] cells treated ± nutlin-3 (10 μM, 48 h). **(E)** Levels of dAMP detected by LC-MS in HT-1080 Control cells treated with DMSO, nutlin-3 (10 μM), or MI-773 (10 μM) for 48 h. Data information: In (C, E), data are mean ± SD of ≥3 independent experiments. *$P < 0.05$, **$P < 0.01$, ***$P < 0.001$, ****$P < 0.0001$) (Student's $t$ test).

ferroptosis is independent of intracellular GSH pools (Yang et al, 2014) (Fig 4F).

Our results suggested that RNR inhibition suppressed ferroptosis by conserving intracellular GSH. To directly test this model, we assayed GSH synthesis and levels by performing metabolic tracing

with uniformly labelled [13]C-serine. Serine can be converted to glycine within the cell, and glycine then incorporated into GSH. Thus, GSH synthesized de novo using [13]C-glycine will be labeled with two heavy carbons and detected as an m+2 isotopologue. Cells were pretreated with Gem for 48 h, then treated with Era2 in

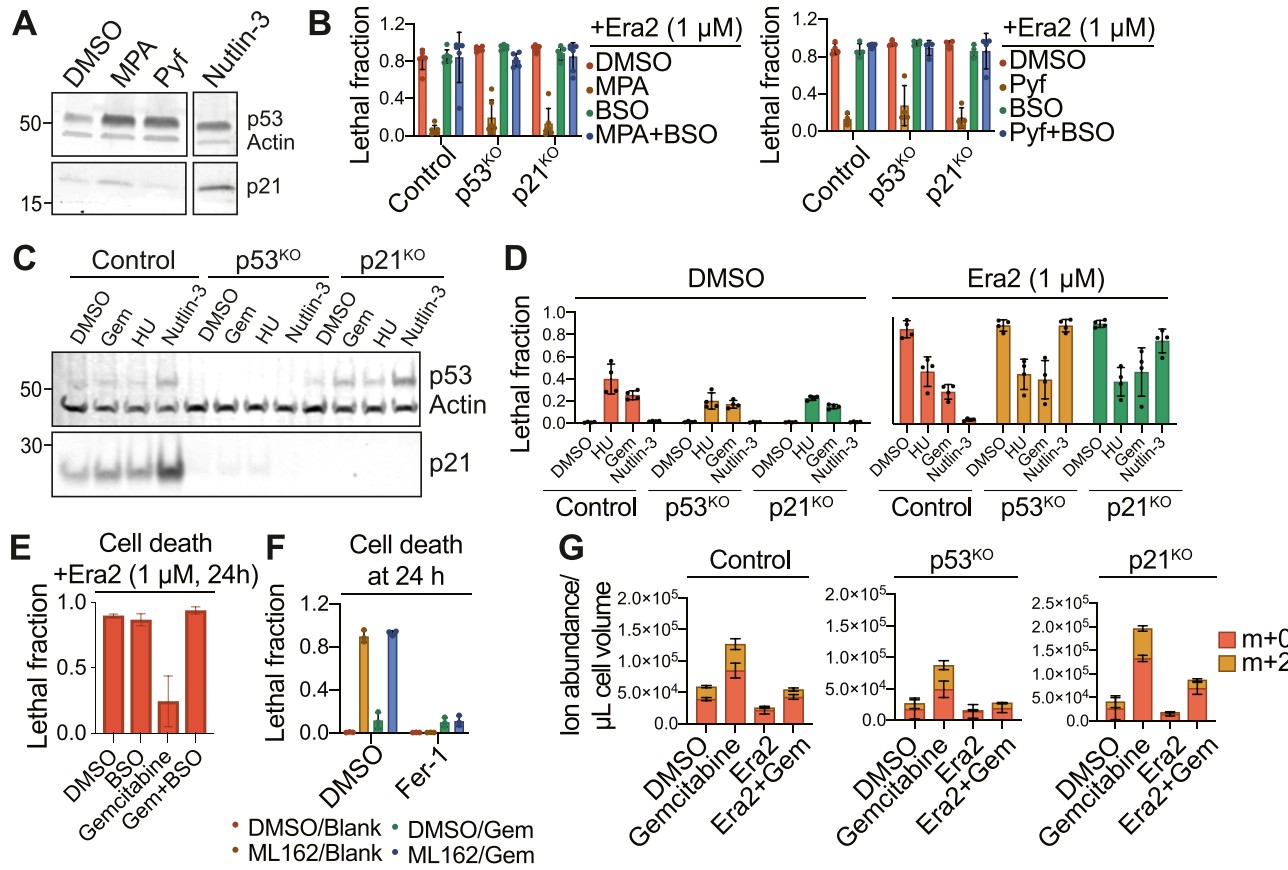

**Figure 4. Pharmacological inhibition of nucleotide metabolism modulates ferroptosis.**
**(A)** Protein expression in HT-1080 Control cells treated with mycophenolic acid (MPA, 5 µM), pyrazofurin (Pyf, 5 µM), or Nutlin-3 (10 µM). **(B)** Cell death in HT-1080[N] Control, p53[KO] and p21[KO] cells pretreated for 48 h with mycophenolic acid (5 µM, left) or pyrazofurin (5 µM, right) ± buthionine sulfoximine (BSO, 100 µM), then treated with erastin2 (Era2, 1 µM, 24 h). **(C)** Protein expression in HT-1080 Control cells treated with DMSO, gemcitabine (Gem, 200 nM), hydroxyurea (HU, 5 mM), or nutlin-3 (10 µM). **(D)** Cell death in HT-1080[N] Control, p53[KO] or p21[KO] cells pretreated with hydroxyurea (5 mM), gemcitabine (200 nM), or nutlin-3 (10 µM), then ± era2 for 24 h. **(E)** Cell death in HT-1080 Control cells pretreated with gemcitabine (Gem, 200 nM) ± buthionine sulfoximine (100 µM), then treated with Era2 (1 µM, 24 h). **(F)** Cell death in HT-1080 Control cells pretreated with ± gemcitabine (200 nM) for 48 h then treated ± ML162 (1 µM) ± ferrostatin-1 (1 µM). **(G)** Metabolic flux assay using LC-MS detection of glutathione in HT-1080 Control, p53[KO] or p21[KO] cells pretreated with DMSO or gemcitabine (200 nM) for 48 h. After pretreatment, medium containing Era2 (1 µM) and U-[13]C Serine was added to cells for 8 h. De novo synthesized glutathione is labeled as an m+2 isotopologue (yellow). Data information: In (B, D, E, F, G), data are mean ± SD of at least three independent experiments.

medium containing [13]C-serine. Pretreatment with Gem increased both the total levels and de novo synthesis of GSH in Control, p53[KO] and p21[KO] cell lines (Fig 4G). Control and p21[KO] cells pretreated with Gem also conserved a larger portion of the existing (m+0) intracellular GSH pool after treatment with erastin2 compared with cells pretreated with DMSO, with weaker effects observed in p53[KO] cells (Fig 4G). These data suggest that RNR inhibition may suppress ferroptosis by conserving intracellular GSH.

### RNR inhibition is sufficient to suppress lipid peroxide accumulation and ferroptosis

To strengthen the results obtained above using pharmacological inhibitors we silenced expression of the essential RNR subunit *RRM1* using siRNA. Silencing of *RRM1* in HT-1080 Control cells decreased the corresponding protein levels and led to compensatory up-regulation of RRM2, indicating that our silencing reagent worked as expected (Fig 5A). Consistent with our pharmacological results, si-*RRM1* treatment resulted in greater conservation of

intracellular GSH m+0 pools after erastin2 treatment compared with si-Control–treated cells, as determined using [13]C-serine metabolic tracing (Fig 5B). Silencing of *RRM1* likewise inhibited the accumulation of lipid peroxides in response to erastin2 treatment, as determined by confocal imaging of the lipid peroxidation probe C11-BODIPY 581/591, whose oxidation is highly correlated with the onset of ferroptosis (Dixon et al, 2012; Magtanong et al, 2019; Conlon et al, 2021) (Fig 5C). Concurrently, *RRM1* silencing was sufficient to suppress ferroptotic cell death in Control cells treated with erastin2 (Fig 5D). This effect of *RRM1* silencing on erastin2-induced ferroptosis was specific: *RRM1* silencing did not inhibit cell death in response to several mechanistically distinct small molecule inducers of apoptosis or other forms of non-apoptotic cell death that are not directly linked to intracellular GSH metabolism (Ko et al, 2019) (Fig S4A). The ability of *RRM1* silencing to inhibit erastin2-induced ferroptosis was partially reverted by co-treatment with BSO, consistent with a GSH-dependent mechanism (Fig S4B). Thus, inhibition of RNR function appeared sufficient to suppress ferroptosis specifically in response to system x$_c^-$ inhibition in Control cells.

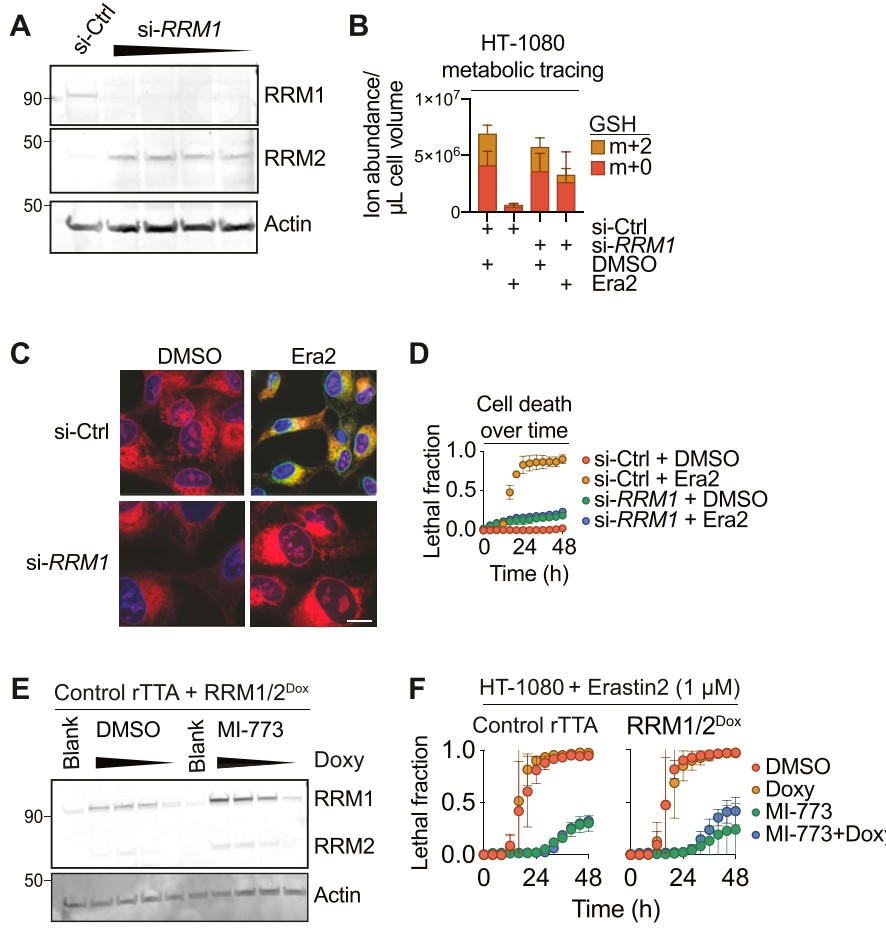

**Figure 5. Ribonucleotide reductase expression modulates glutathione metabolism.**
**(A)** Protein expression in HT-1080 Control cells treated with non-targeting siRNAs (si-Ctrl), or an siRNA pool targeting *RRM1* (si-*RRM1*). Cells were pretreated with siRNA at doses ranging from 1 to 0.125 nM for 48 h before cell harvest. **(B)** Metabolic tracing of HT-1080 Control cells pretreated with si-Ctrl or si-*RRM1* for 48 h, then treated ± erastin2 (Era2) (1 μM, 8 h) in the presence of $^{13}$C-serine. De novo synthesized glutathione is represented as an m+2 isotopologue (yellow). **(C)** Confocal C11-BODIPY 581/591 imaging of lipid peroxidation in HT-1080 Control cells pretreated with si-Ctrl or si-*RRM1* for 48 h, then treated ± Era2 (1 μM, 9 h). Note: this time point was selected because it allowed for harvest of cells just prior to the onset of frank membrane permeabilization. Scale bar = 20 μm. **(D)** Cell death in HT-1080[N] Control cells pretreated with si-Ctrl or si-*RRM1* for 48 h, then treated ± Era2 (1 μM). **(E)** Protein expression in HT-1080[rTTA] cells stably expressing doxycycline (Doxy)-inducible RRM1/2 (RRM1/2[Doxy]), pretreated with medium alone (Blank), or medium containing 10-fold dilutions of Doxy from 1 μg/ml down to 1 ng/ml. **(F)** Cell death in HT-1080 cells pretreated ± MI-773 (10 μM) ± Doxy (100 ng/ml) for 48 h, then treated with Era2. Data information: In (B, D, F), data are mean ± SD of at least three independent experiments.

Next, we tested whether overexpression of RNR alone could restore normal ferroptosis sensitivity in cells where the p53/p21 pathway was activated. Towards this end, we generated a Dox-inducible overexpression construct driving the expression of *RRM1* and *RRM2* joined with a self-cleaving P2A cassette (HT-1080 RNR[Dox]). Dox treatment of HT-1080 RNR[Dox] cells resulted in expression of both RRM1 and RRM2 (Fig 5E). Optimization experiments suggested that nutlin-3 may weakly activate the expression of Dox-inducible cassettes non-specifically. Thus, in this experiment, we used the structurally distinct HDM2 inhibitor MI-773, which like nutlin-3 potently suppressed ferroptosis (Fig 1C). MI-773 pretreated cells were potently protected from erastin2-induced ferroptosis (Fig 5F). However, Dox-induced expression of RRM1 and RRM2 alone was not sufficient to revert this protective effect, possibly because of coordinate down-regulation of multiple nucleotide metabolic pathway genes in response to p53 stabilization (Fig 3B).

# Discussion

Nucleotide synthesis is a demanding task that consumes numerous metabolic resources. Of special relevance to our study, RNR consumes reducing equivalents supplied by GSH, as well as thioredoxin, in mammalian cells (Zahedi Avval & Holmgren, 2009; Sengupta et al, 2019). GSH and thioredoxin also inhibit ferroptosis (Yang et al, 2014; Llabani et al, 2019). RNR-dependent nucleotide synthesis may therefore compete with GPX4-dependent lipid hydroperoxide reduction for use of the same co-substrate, namely GSH. We find that blockade of de novo GSH synthesis eliminates the protective effects of genetic or pharmacological RNR inhibition on ferroptosis. Based on these results, we propose that inhibition of RNR (or upstream enzymes) leads to accumulation of GSH which can be re-purposed for use by GPX4 to prolong the inhibition of ferroptosis under conditions where new GSH synthesis is otherwise impossible (Fig 6). An important caveat of this model is that it is difficult to disentangle the effects of direct RNR inhibition from more global cell cycle arrest. For example, p53 stabilization results in down-regulation of numerous metabolic genes, and we observe that overexpression of RNR subunits alone is not sufficient to re-sensitize arrested cells to ferroptosis, perhaps due to the absence of upstream precursors that would allow RNR to operate at full capacity. Regardless, the possibility that down-regulation of other redox-dependent genes or processes upon p53 stabilization, or in response to other conditions leading to cell cycle arrest, can promote GSH accumulation and inhibition of ferroptosis cannot be excluded. How cells may normally prioritize limited intracellular GSH pools for use in nucleotide synthesis versus lipid hydroperoxide reduction is an important direction for future investigation.

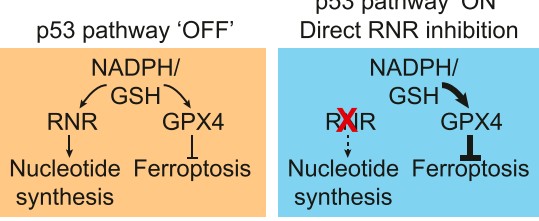

**p53 pathway 'OFF'**

NADPH/GSH

RNR ⟍ ⟋ GPX4

Nucleotide synthesis | Ferroptosis

**p53 pathway 'ON'**
**Direct RNR inhibition**

NADPH/GSH

R✗R ⟍ ⟋ GPX4

Nucleotide synthesis | Ferroptosis

**Figure 6. A model of p53–p21 activity in ferroptosis suppression.**
Stabilization of wild-type p53 leads to reduction of ribonucleotide reductase expression and function, and conservation of intracellular glutathione and potentially other reducing agents (e.g., NADPH), which can then be re-directed towards the inhibition of ferroptosis. Note: p53 may down-regulate *RRM1* and *RRM2* in a partly p21-independent manner.

We propose that stabilization of wild-type p53 and induction of p21 can link down-regulation of RNR-associated gene expression to protect against ferroptosis. Such a model would be consonant with, and elaborate upon, previous findings that wild-type p53 expression can protect against serine deprivation-associated oxidative stress (Maddocks et al, 2013). Of note, repression of RRM1/2 upon p53 stabilization was partly independent of p21. This may explain why p21 overexpression alone was not sufficient to fully inhibit ferroptosis in our hands: p53 stabilization is needed together with p21 induction for complete inhibition of RNR complex expression and function. However, others have observed that p21 expression alone can be sufficient to inhibit ferroptosis (Venkatesh et al, 2020b). It is possible that these differences are explained by cell type–specific differences, or the ability of p21 to regulate redox homeostasis via alternative means in some contexts (Chen et al, 2009). Regardless, these results point towards a meaningful role of p21 in regulating cell fate that is only partly dependent on p53.

In the context of cancer therapy, our results suggest that stimulation of nucleotide synthesis in combination with GSH depletion may be especially toxic. By contrast, the effects of anti-cancer agents that seek to induce ferroptosis via GSH depletion (e.g., cyst(e)inase, [Cramer et al, 2017]) could be blunted if combined with chemotherapeutics such as gemcitabine and HU, which inhibit the consumption of GSH by nucleotide metabolic processes and also slow DNA replication by blocking elongation (Heinemann et al, 1990; Plunkett et al, 1995). Indeed, other conditions that slow intracellular metabolism, such as inhibition of mTOR signaling or deprivation of certain amino acids, also appear to increase resistance to ferroptosis through GSH-conserving mechanisms that may be unrelated to nucleotide synthesis (Conlon et al, 2021). Collectively, these studies suggest somewhat paradoxically that to maximize the induction of ferroptosis by GSH depletion it may be necessary to ensure that cancer cells are otherwise as metabolically active as possible.

# Materials and Methods

### Cell lines and culture conditions

Cell viability studies used human polyclonal cell lines expressing nuclear-localized mKate2. Polyclonal nuclear-localized mKate2-expressing HT-1080 cells (denoted HT-1080[N]), U-2 OS[N], H1299[N],

and Caki-1[N] cell lines were described previously (Forcina et al, 2017; Conlon et al, 2021). HT-1080[N] *TP53* (p53) and *CDKN1A* (p21) gene disrupted (KO) cell lines generated using CRISPR-Cas9 genome editing were described previously (Tarangelo et al, 2018). U-2 OS[N] cells stably transduced with an empty vector (CMV-Empty) or CMV-MRP1 were described previously (Cao et al, 2019). Rho zero ($\rho^0$) HT-1080 cells were generated as previously described (Dixon et al, 2012). Briefly, HT-1080 cells were cultured in media containing 10 ng/ml ethidium bromide and 50 $\mu$g/ml uridine for 14 passages. Uridine was maintained in the media for all experiments and was also supplemented into the medium of control $\rho^+$ cells.

HT-1080 and 293T cells were cultured in DMEM Hi-glucose medium (Cat. no. MT-10-013-CV; Corning Life Science) supplemented with 1% nonessential amino acids (Cat. no. 11140-050; Life Technologies), 10% FBS (Cat. no. 26140-079; Gibco), and 0.5 U/ml Pen/Strep (P/S, Cat. no. 15070-063; Gibco). H1299 cells were cultured in DMEM Hi-glucose medium, with 10% FBS and 0.5 U/ml P/S only. U-2 OS and Caki-1 cells were cultured in McCoy's 5A media (Cat. no. MT-10-050-CV; Corning Life Science) supplemented with 10% FBS and 0.5 U/ml P/S. All media were filtered through a 0.22 $\mu$M PES filter (Genesee Scientific) before use. All cell lines were grown at 37°C with 5% $CO_2$ in humidified tissue culture incubators (Thermo Fisher Scientific).

### Chemicals and reagents

Erastin2 and ML162 were synthesized at Acme Bioscience. Nutlin-3 (Cat. no. S1061), nutlin-3b (Cat. no. S8065), gemcitabine HCL (Cat. no. S1149), and MI-773 (Cat. no. S7649) were purchased from Selleck Chemicals. Pyrazofurin (Cat. no. SML1502), HU (Cat. no. H8627), MPA (Cat. no. M5255), ferrostatin-1 (Cat. no. SML0583), Doxycycline hyclate (Cat. no. D9891), and L-BSO (Cat. no. B2515) were purchased from Sigma-Aldrich. BSO and gemcitabine HCL were dissolved directly into cell media. C11-BODIPY 581/591 was prepared as a stock solution in methanol and stored before use at −20°C. All other compounds were prepared as stock solutions in DMSO and stored before use at −20°C.

### Cell viability experiments

All experiments examining the effects of p53 activation or RNR inhibition on cell death used a pretreatment phase (24 or 48 h). On day 1, cells were seeded into clear-bottom 96-well assay plates (Cat. no. 07-200-588; Thermo Fisher Scientific). Cell seeding was optimized to achieve approximately equal densities of cells in each well, accounting for the cytostatic effects of some compounds during the pretreatment phase. Cells were seeded at the following concentrations. HT-1080 Control[N]-treated with DMSO or nutlin-3b: 1,000–1,500, MDM2 inhibitors: 3,000–4,000, or gemcitabine: 3,000–4,000; HT-1080[N] p53[KO] treated with DMSO: 750–1,000, MDM2 inhibitors: 750–1,000, or gemcitabine: 2,000; HT-1080[N] p21[KO] treated with DMSO: 2,000–3,000, MDM2 inhibitors: 4,000–5,000, gemcitabine: 4,000. Caki-1[N] or U-2 OS[N] cells treated with DMSO: 1,500, nutlin-3: 3,000, gemcitabine: 3,000; H1299[N] treated with DMSO: 1,000, nutlin-3: 1,000 or gemcitabine: 2,000. The pretreatment phase began on day 2 when medium was replaced with medium containing DMSO or a small molecule inhibitor (e.g., nutlin-3 [10 $\mu$M] or gemcitabine HCL [200 nM]). During the

pretreatment phase, cell death was not monitored. On day 4, after 48 h of pretreatment, the medium was removed and replaced with medium containing lethal compounds or other modulating compounds (such as BSO [100 $\mu$M]). DMSO or pretreatment compounds were maintained in the medium during the observed treatment phase. During the treatment phase, cell death was observed and quantified using STACK as previously described (Forcina et al, 2017).

## Cell death kinetics quantification

Analysis of cell death kinetics was performed using STACK analysis (Forcina et al, 2017). After pretreatment, infection, or transfection, cell medium was exchanged for medium containing lethal compounds and 0.022 $\mu$M of the viability dye SYTOX Green (SG, Cat. no. S7020; Life Technologies). Cells were then imaged at 4 h intervals for ≥48 h using the Essen IncuCyte ZOOM live-cell analysis system (Essen BioSciences). Live cells were quantified by counting mKate2-positive cells, and dead cells were quantified by counting SG-positive cells. All quantification was performed automatically using the IncuCyte ZOOM Live-Cell Analysis System software (Essen BioSciences). Lethal fraction scores were calculated at each time point as previously described, with two modifications (Forcina et al, 2017): (i) because long-dead cells can release SG, the maximum number of SG positive counts from the start of the time course was used in these calculations; (ii) SG/mKate2 double-positive cells were considered to be in the process of death, and thus counts of double positive cells were subtracted from the counts of live cells. Lag exponential death curves were fitted to lethal fraction data using Prism 8 (GraphPad), as described (Forcina et al, 2017). The parameter of time to death onset, or $D_O$ was computed from lag exponential death curve fits, as previously described (Forcina et al, 2017).

## Immunoblotting

Cells seeded in six-well dishes were washed twice with HBSS (Cat. no. 14025-134; Life Technologies) and collected using a manual cell lifter. Cell pellets were collected and lysed in 9 M urea or RIPA buffer (10 mM Tris pH 7.5, 150 mM NaCl, 1 mM EDTA, 0.5% Na deoxycholate, 0.1% SDS, and 1% Triton X). Lysates were sonicated 10 times with 1-s pulses at 80% amplitude on a Fisher Scientific Model 120 Sonic Dismembrator (Thermo Fisher Scientific), then centrifuged at 16,000$g$ at room temperature (for 9 M urea) or at 4°C (for RIPA lysis) for 15–20 min. Lysates were quantified by BCA assay (Cat. no. PI23224; Thermo Fisher Scientific). Samples were prepared with NuPage 10× Reducing Agent (Cat. no. NP0009; Thermo Fisher Scientific) and NuPage LDS 4× Sample Buffer (Cat. no. NP0007; Thermo Fisher Scientific). Samples were incubated at 70°C for 10 min and run on pre-cast NuPage SDS 4–12% gradient gels in NuPage MES Running Buffer (Cat. no. NP0002; Thermo Fisher Scientific). Gels were transferred to nitrocellulose membranes using an iBlot Dry Blotting System (Cat. no. IB21001; Thermo Fisher Scientific), blocked for ≥1 h in 50% Odyssey PBS (Cat. no. 927-40010; LI-COR)/50% diH$_2$O, and probed with primary antibodies either overnight at 4°C or at room temperature or 1–3 h. Primary antibodies used were against actin (Cat. no. SC-1616, I-19 or Cat. no. sc-47778, C4; Santa Cruz Biotechnology, 1:1,000), human p53 (Cat. no. SC-126, DO-1; Santa Cruz

Biotechnology, 1:1,000), p21 C′-terminus (Cat. no. 2947, 12D1; Cell Signaling Technology, 1:1,000), p21 N′-terminus (Cat. no. ab80633, CP74; Abcam, 1:500), RRM1 (Cat. no. CST3388; Cell Signaling Technologies, 1:500), and RRM2 (Cat. no. 65939; Cell Signaling Technologies, 1:500). Membranes were washed three times for 10 min each in Tris-buffered saline (Cat. no. 0788; ISC BioExpress) with 0.1% Tween 20 (TBST) at room temperature with rocking. Membranes were probed with secondary antibodies (Donkey anti-goat-680 Cat. no. 926-68024, Donkey anti-goat-800 Cat. no. 926-32214, Donkey anti-rabbit-680 Cat. no. 926-68023, Donkey anti-rabbit-800 Cat. no. 926-32213, Donkey anti-mouse-680 Cat. no. 926-68022, Donkey anti-mouse-800 Cat. no. 926-32212; LI-COR) in 50% Odyssey Buffer (Cat. no. 927-40100; LI-COR)/50% diH$_2$O with 0.1% SDS and 0.4% Tween 20 for 1 h at room temperature with rocking. Membranes were washed three times for 10 min each in TBST, then imaged on an LI-COR Odyssey CLx imager.

## Inducible gene expression and cloning

HT-1080 cell lines expressing tetracycline-inducible FL or truncated forms of p21[20] were generated as follows. gBlocks containing a FLAG tag followed by the full coding sequence of human *CDKN1A*, the N′ terminus only (amino acids 1–90), or the C′ terminus (amino acids 87–164) were synthesized by IDT DNA and cloned into lentiviral vectors containing a Tet-inducible promoter (pLenti-CMV-TRE3G-Puro, a gift of Jan Carette) by Gibson assembly. HT-1080[N] cells were transduced with lentiviruses carrying the reverse tetracycline-controlled transactivator 3 under a CMV promoter (CMV-rtTA3) (Cat. no. w756-1; Addgene). HT-1080[N-rtTA3] cell lines were selected in 10 $\mu$g/ml blasticidin (Cat. no. A1113902; Thermo Fisher Scientific) for 5 d. These cell lines were then transduced with lentiviruses carrying FL or truncated FLAG-tagged p21 coding sequences under a Tet-inducible promoter. Cell lines were selected with 1 $\mu$g/ml puromycin for 3 d.

To overexpress RNR, the following construct was synthesized as gBlocks and assembled using Gibson assembly (Master Mix Cat. no. E2611S; New England Biolabs). The FLAG-tagged coding sequence of *RRM1* was joined via a P2A cassette to the 6x-His tagged coding sequence of *RRM2*. This construct was cloned into the pLenti-CMV-TRE3G vector (pLenti-CMV-TRE3G-FLAG-RNR-P2A-HIS-RRM2). HT-1080[N-rtTA3] cell lines were infected with viruses carrying pLenti-CMV-TRE3G-FLAG-RNR-P2A-HIS-RRM2 and selected using 1 $\mu$g/ml puromycin for 3 d. To induce gene expression in tetracycline-inducible cell lines, cells were treated with 1 ng–1 $\mu$g Dox for 24–48 h. Gene expression was confirmed using immunoblotting as described above.

## Lentivirus production

To generate lentiviruses, 293T cells were seeded at a density of 0.5 × 10$^6$ cells in a six-well dish the day before transfection. Cells were transfected the next day with 1,000 ng of plasmid DNA plus 250 ng pMD2.G (Cat. no. 12259; Addgene) and 750 ng psPax2 (Cat. no. 12260; Addgene) second generation lentiviral packaging plasmids. Transfection was performed using 3 $\mu$l PolyJet (Cat. no. SL100688; SignaGen Laboratories) transfection reagent diluted in plain DMEM to a final volume of 100 $\mu$l and incubated for 15 min at room temperature. The transfection mixture was added drop-by-drop to wells and incubated

for 24 h. After 24 h, the cell supernatant was removed and replaced with 1.5–2 ml fresh complete DMEM. After 8 h, virus-containing medium was harvested and replaced with 1.5–2 ml fresh complete DMEM. The following day, cell medium was harvested twice more at an interval of >8 h, for a total of three harvests. Virus-containing medium was filtered using a 0.45-$\mu$M PVDF (Cat. no. SLHV033RS; Millex) syringe-driven filter unit and frozen at –80°C until use. Relative viral titer was determined by infecting cells with serially diluted volumes of virus-containing medium and selecting using appropriate selection reagents.

## siRNA gene knockdown

Cells were reverse-transfected with an ON-TARGET Plus siRNA SMARTPool targeting human *RRM1* (targeting NM_001033.5, NM_001318064.1, NM_001318065.1, NM_001330193.1, Cat. no. L-004270-00-0005; Horizon). AllStars Negative Control siRNA (Cat. no. 1027280; QIAGEN) was used as a negative control. AllStars Hs Cell Death siRNA (Cat. no. 1027298; QIAGEN) was used as a positive control for transfection efficiency in all experiments. Transfection mixes were prepared using 0.25 nM siRNA and 2.5 $\mu$l Lipofectamine RNAiMAX Transfection Reagent (Cat. no. 13778075; Life Technologies) to a total volume of 500 $\mu$l transfection mixture in Opti-MEM Reduced Serum Media (Cat. no. 31985-062; Life Technologies). 250 $\mu$l Opti-MEM was pipetted into empty six well dishes and siRNA was added directly to the wells. 250 $\mu$l of Lipofectamine mixture was added to the siRNA mixture. The combined mixture was swirled and incubated 15 min at room temperature. HT-1080 cells were seeded in a six-well at the following densities per well. HT-1080 Control: siControl, 0.5–1 × 10$^5$; si-*RRM1*, 1.5–2 × 10$^5$. HT-1080 p53$^{KO}$ and p21$^{KO}$ cell lines were seeded in a 12-well dish at the following densities. HT-1080 p53$^{KO}$: siControl, 0.15 × 10$^5$; si-*RRM1*, 0.7 × 10$^5$. HT-1080 p21$^{KO}$: siControl, 0.3 × 10$^5$; si-*RRM1*, 1 × 10$^5$. HT-1080 cells were seeded in a six-well at the following densities per well: siControl, 0.6 × 10$^5$; si*RRM1*, 1.5 × 10$^5$. Cells were added in a volume of 1.5 ml, for a total well volume of 2 ml. Cells were incubated for 48 h before assay.

## C11-BODIPY 581/591 imaging

HT-1080 cells were seeded on glass coverslips in six-well dishes at the following densities: Blank Media, 0.3 × 10$^5$; Gemcitabine, 1 × 10$^5$; siControl, 0.5 × 10$^5$; si-*RRM1*, 2 × 10$^5$. For the *RRM1* knockdown condition, reverse transfection reagent was prepared on top of glass coverslips in six-well dishes using either non-targeting control siRNAs, or siRNA pools targeting human *RRM1*, as described above. On day 2, the medium was removed and replaced with fresh complete medium. For the Gemcitabine treatment condition, cells were seeded on day 1 as described above. On day 2, cells were pretreated with or without Gemcitabine HCL (200 nM). For both conditions, on day 4, cell medium was removed and replaced with medium containing DMSO or 1 $\mu$M Erastin2. Cells were incubated at 37°C for 9.5 h, before the onset of cell death, after which point the cell medium was removed and cells were washed with HBSS. C11-BODIPY 581/591 was prepared as a 5 $\mu$M working solution in HBSS and added to cells along with 1 $\mu$g/ml Hoechst 33258 stain (Cat. no. H3569; Thermo Fisher Scientific). Cells were incubated for 10 min at 37°C, then the C11-BODIPY mixture was removed and cells were washed in 1 ml of fresh HBSS. To prepare slides for imaging,

25 $\mu$l HBSS was pipetted onto a microscope slide lined with parafilm. Coverslips with stained cells were lifted out of plates using a needle tip and inverted onto prepared slides. Slides were sealed with melted Vaseline and immediately imaged on a Zeiss Observer Z1 confocal microscope. Images were processed in ImageJ. Brightness for the green (oxidized BODIPY) channel was auto-scaled using images from cells treated with erastin2. Brightness for all images was auto-adjusted based on imaged with the brightest signal (i.e., brightness for the red (non-oxidized BODIPY) channel was auto-scaled using images from cells treated with DMSO, whereas brightness for the green (oxidized BODIPY) channel was auto-scaled using images of cells treated with erastin2).

## Image analysis

Image processing was performed in ImageJ (version 1.50i) or Adobe Photoshop CS6 (Adobe Systems).

## Reverse transcription and quantitative PCR

Cells were seeded in six-well dishes and treated with desired reagents. After treatment, cells were washed twice with HBSS and collected using a cell lifter. Pellets were centrifuged at room temperature for 2 min at 1,000$g$. Supernatants were removed and lysed for RNA extraction using a Qiashredder extraction column (Cat. no. 79654; QIAGEN) and the RNeasy Plus RNA Extraction Kit (Cat. no. 74134; QIAGEN). Reverse transcription to generate cDNA was performed using TaqMan Reverse Transcriptase Kit according to the manufacturer's instructions (Cat. no. N8080234; TaqMan). Quantitative PCRs were prepared using SYBR Green Master Mix (Cat. no. 4367659; Life Technologies) and run on an Applied Biosystems QuantStudio 3 real-time PCR machine (Thermo Fisher Scientific). Relative transcript levels were calculated using the $\Delta\Delta$CT method and normalized to the *ACTB* gene. Primer sequences are as follows: *ACTB*, forward: ATCCGCCGCCCGTCCACA, reverse: ACCATCACGCCCTGGTGCCT; *CKDN1A*, forward: CACCGAGACACCACTGGAGG, reverse: GAGAA-GATCAGCCGGCGTTT; *MDM2*, forward: GAATCATCGGACTCAGGTA-CATC, reverse: TCTGTCTCACTAATTGCTCTCCT; *RRM1*, forward: CAGTGATGTGATGGAAGA, reverse: CTCGGTCATAGATAATAGCA; *RRM2*, forward: AGACTTATGCTGGAACT, reverse: TCTGATACTCGCCTACTC; *ND1* forward: CCACATCTACCATCACCCTC, reverse: TTCATAGTAGAAGAGCGATGGT; *ND2*, forward: GACATCCGGCCTGCTTCTT, reverse: TACGTTTAGTGAGG-GAGAGATTTGG; *ND5*, forward: TTCAAACTAGACTACTTCTCCATAATATTC ATC, reverse: TTGGGTCTGAGTTTATATATCACAGTGA.

## RNA sequencing

HT-1080 cells were seeded in six-well dishes at the following densities. Control HT-1080s: DMSO, 0.5 × 10$^5$, Nutlin-3, 1.5 × 10$^5$. HT-1080 p53$^{KO}$: DMSO, 0.3 × 10$^5$, Nutlin-3, ×10$^5$. HT-1080 p21$^{KO}$: DMSO, 0.6 × 10$^5$, Nutlin-3, 1.5 × 10$^5$. The cells were treated for 48 h with DMSO or 10 $\mu$M nutlin-3. The cells were then washed with ice-cold HBSS and detached with a cell lifter on ice. Detached cells were pelleted by centrifugation at 1,000$g$ for 2 min. The supernatant was removed and cell pellets were immediately lysed and RNA was extracted using a Qiashredder extraction column (Cat. no. 79654; QIAGEN) and the RNeasy Plus RNA Extraction Kit (Cat. no. 74134; QIAGEN). Two

biological replicates were collected for each sample. Purity, concentration, and integrity of RNA was assessed by NanoDrop Spectrophotometer (Thermo Fisher Scientific) and by Eukaryote Total RNA Nano chip analysis on a Bioanalyzer (Agilent Technologies) at the Stanford Protein and Nucleic Acid Facility. The RNA Integrity Number for each sample met or exceeded the minimum threshold of 6.8, and NanoDrop 260/280 and 260/230 ratios achieved a threshold of at least 1.95. Following quality control steps, samples were shipped on dry ice to Novogene for library generation and 20M read PE150 sequencing on an Illumina HiSeq 4000 Platform. Data cleanup was performed by excluding reads with: adaptor contamination; >10% of indeterminate bases, or >50% bases with a quality score ≤5. The remaining reads (≥98% of all reads across all conditions) were aligned to the hg19 human reference genome using STAR. At least 91% of clean reads were mapped across all conditions. Pearson correlation between biological replicates was $R^2 \geq 0.96$ for all samples. All FPKM counts for each condition are accessible online at Mendeley Data (http://dx.doi.org/10.17632/gbvp7h5h8j.1). Differential gene expression was calculated using DESeq 1.10.1. We selected all protein-coding genes with read counts above 0.5 FPKM in all conditions and with significant alterations (adjusted $P$-value < 0.05).

### Steady-state metabolite analysis

The levels of dAMP were measured using liquid chromatography coupled to mass spectrometry. Briefly, HT-1080 cells were seeded into 15-cm dishes at a density of $1 \times 10^6$ cells/dish, then the next day treated ± nutlin-3 (10 $\mu$M). 48 h later, cells were harvested by trypsinization, pelleted (1,500$g$ for 5 min), and stored at –80°C. Four biological replicates were collected and analyzed by Metabolon as described previously (Skouta et al, 2014).

### Metabolic tracing analysis

For these experiments, cells were cultured in RPMI 1640 medium (Cat. no. SH30027FS; Thermo Fisher Scientific) containing 10% FBS and 1× P/S. HT-1080[N] cells were seeded in six-well dishes at the following densities per well. Control HT-1080[N]: DMSO, $0.3 \times 10^5$, gemcitabine, $1.5$–$2 \times 10^5$. HT-1080[N] p53[KO]: DMSO, $0.25 \times 10^5$, gemcitabine, $0.75 \times 10^5$. HT-1080[N] p21[KO]: DMSO, $0.4 \times 10^5$, gemcitabine, $1.5 \times 10^5$. The following day, cells were pretreated with blank medium or gemcitabine HCL (200 nM) for 48 h in RPMI 1640 medium. For iterations of this experiment ± gemcitabine (200 nM) conditions were maintained from the pretreatment phase. Cells were incubated in [U-$^{13}$C] L-serine medium for 8 h. Unlabeled controls were incubated for 8 h in standard RPMI 1640 medium. In a separate iteration of this experiment, HT-1080 cells were reverse-transfected with non-targeting (control) siRNAs or siRNAs targeting RRM1 and seeded as described above. After 48 h, medium was removed and replaced with RPMI medium lacking glucose, serine, and glycine (Cat. no. R9660-02; Teknova) supplemented with 30 mg/liter [U-$^{13}$C] L-serine (Cat. no. 604887; Sigma-Aldrich) and 2,000 mg/liter D-glucose (Cat. no. G54000; Sigma-Aldrich) ± erastin2 (1 $\mu$M). Before collection, cells were imaged on the IncuCyte ZOOM and total cell number of the well was computed based on counts of the total mKate2$^+$ objects per area of viewing field. After the treatment phase, cells were

washed twice in ice-cold HBSS, and fixed in 80% cold LC/MS grade acetonitrile (Cat. no. A955-500; Thermo Fisher Scientific) for 5 min on ice. Cells were collected using a cell lifter, centrifuged at 14,000$g$ for 10 min at 4°C, and the resulting supernatant was removed to glass vials and stored at –80°C before analysis. The remaining pellets were frozen at –80°C. Because of the effect of gemcitabine and RRM1 knockdown on cell size, ion abundance was normalized to total cell volumes. To quantify cell volume, cells were seeded and treated exactly as described above. After appropriate pretreatment phases, cells were trypsinized, centrifuged for 5 min at 200$g$, and resuspended in 500 $\mu$l ISOTON Diluent (Cat. no. 12754878; Thermo Fisher Scientific). Cell suspensions were diluted 250 $\mu$l into 10 ml of ISOTON diluent and cell size was measured on a Coulter Z2 Cell Analyzer (Cat. no. 844 80 11; Beckman Coulter). Median cell volume was used for calculations. Median cell volume was multiplied by the number of cells counted before collection to yield a total cell volume for each sample. Ion abundances were normalized to cell volumes.

For separation of polar metabolites, normal-phase chromatography was performed with a Luna 5 $\mu$m NH2 100 Å LC column (00B-4378-E0; Phenomenex). Mobile phases were as follows: Buffer A, acetonitrile; Buffer B, 95:5 water/acetonitrile with 0.2% ammonium hydroxide and 50 mM ammonium acetate for negative ionization mode. The flow rate for each run started 100% B for 2 min at 0.2 ml/min, followed by a gradient starting at 20% A/80% B changing linearly to 70% A/30% B over the course of 1 min at 0.7 ml/min, followed by 70% A/30% B for 5 min at 0.7 ml/min, and finally to 100% B over a course of 5 min. The overall runtime was 13 min and the injection volume was 20 $\mu$l.

MS analysis was performed with an electrospray ionization source on an Agilent 6545 qTOF LC/MS at the Metabolic Chemistry Analysis Center at Stanford ChEM-H. For QTOF acquisition parameters, the mass range was set from 50 to 1,000 m/z with an acquisition rate of five spectra/second and time of 250 ms/spectrum. For Dual AJS electrospray ionization source parameters, the drying gas temperature was set to 250°C with a flow rate of 12 liter/min, and the nebulizer pressure was 20 $\psi$. The sheath gas temperature was set to 300°C with a flow rate of 12 liter/min. The capillary voltage was set to 3,500 V and the fragmentor voltage was set to 100 V. Isotopologues extraction was performed in Agilent Profinder B.08.00 (Agilent technologies). Retention time of each metabolite was determined by authentic standards. The mass tolerance was set to ±15 ppm and RT tolerance was ±0.2 min. GSH (formula $C_{10}H_{17}N_3O_6S$, METLIN: 44, HMP: HMDB00125, KEGG C00051) was detected with a mass of 307.084 and a retention time of 8.78 min.

### Seahorse assay

HT-1080 cells were seeded in Seahorse XFp six-well miniplates at the following densities. HT-1080 Control: DMSO, 750 cells; Nutlin-3, 2,000 cells; Palbociclib, 1,500 cells. HT-1080 p53[KO]: DMSO, 500 cells; Nutlin-3, 500 cells; Palbociclib, 500 cells. HT-1080 p21[KO]: DMSO, 1,000 cells; Nutlin-3, 2,000 cells; Palbociclib, 1,500 cells. Cells were pretreated with small molecules in Seahorse XFp plates for 48 h before assay. For $\rho^0$ validation, 8000 HT-1080 Control $\rho^+$ and 12,000 HT-1080 Control $\rho^0$ cells were seeded into XFp six-well mini plates and assayed the next day. XFp Mitochondrial and Glycolytic Stress Tests

were performed per the manufacturer's instructions. All data were normalized to cell number as determined by counts of mKate2[+] cells as imaged using the IncuCyte before the start of the assay.

### Graphing and figure assembly

Graphing and statistical analyses were performed using GraphPad Prism 8. The Results and individual figure legends contain additional statistical details. Figures were constructed in Adobe Illustrator.

### Data analysis

Data analysis was performed using Microsoft Excel and GraphPad Prism 8. Data are represented as mean ± SD or ± 95% confidence interval unless otherwise noted.

## Data Availability

RNA sequencing data are accessible online at Mendeley Data (http://dx.doi.org/10.17632/gbvp7h5h8j.1).

## Supplementary Information

## Acknowledgements

We thank Tim Stearns for help with confocal imaging, Edo Biluar for reading the manuscript and assisting with experiments, and Julien Sage and Jan Carette for providing reagents. This work was supported by a National Institutes of Health (NIH)/National Cancer Institute F99/K00 Predoctoral to Postdoctoral Transition Award to A Tarangelo (1F99CA234650-01) and awards from the NIH (1R01GM122923) and American Cancer Society (RSG-21-017-01) to SJ Dixon.

### Author Contributions

A Tarangelo: conceptualization, formal analysis, investigation, methodology, and writing—original draft, review, and editing.
J Rodencal: investigation and methodology.
JT Kim: investigation, methodology, and writing—review and editing.
L Magtanong: investigation and methodology.
JZ Long: resources, supervision, funding acquisition, methodology, and writing—review and editing.
SJ Dixon: conceptualization, formal analysis, supervision, funding acquisition, methodology, project administration, and writing—original draft, review, and editing.

### Conflict of Interest Statement

SJ Dixon is a co-founder of Prothegen Inc., member of the scientific advisory boards of Ferro Therapeutics and Hillstream Biopharma, and holds patents related to ferroptosis.

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
