## [Reviewer comments · Life Science Alliance]

Life Science Alliance

Nucleotide Biosynthesis Links Glutathione Metabolism to Ferroptosis Sensitivity

Amy Tarangelo, Jason Rodencal, Joon Kim, Leslie Magtanong, Jonathon Long, and Scott Dixon

DOI: <https://doi.org/10.26508/lsa.202101157>

Corresponding author(s): Scott Dixon, Stanford University

Review Timeline:

Submission Date:	2021-07-13
Editorial Decision:	2021-08-20
Revision Received:	2021-12-08
Editorial Decision:	2021-12-13
Revision Received:	2022-01-06
Accepted:	2022-01-11

Scientific Editor: Novella Guidi

Transaction Report:

August 20, 2021

Re: Life Science Alliance manuscript #LSA-2021-01157

Dr. Scott J Dixon
Stanford University
Department of Biology
337 Campus Dr., Room 104
Lokey Chemistry & Biology Building
Stanford, CA 94305

Dear Dr. Dixon,

Thank you for submitting your manuscript entitled "Nucleotide Biosynthesis Links Glutathione Metabolism to Ferroptosis Sensitivity" to Life Science Alliance. The manuscript was assessed by expert reviewers, whose comments are appended to this letter. Both Reviewers feel like the study is well-designed, the experiments are rigorously conducted and that the results will be of interest to the ferroptosis community and more broadly to the cancer community. However, they do raise some concerns with the main one being the lack of RNA seq data deposition and analysis. Therefore, we encourage you to provide analyzed RNAseq data presented in Figure 3 as a supplementary dataset or indicate that these data have been publicly deposited. Moreover Reviewer 1 points to the importance of investigating the extent of the changes in lipid peroxide levels to strengthen the conclusions made using C11 BODIPY and suggests the authors to use other cell lines to study ferroptosis. Reviewer 2 would like the authors to discuss cell-cycle dependent effects to disentangle inhibition of replication from inhibition of nucleotide synthesis and run a rescue experiment in which RNR1 suppression rescues ferroptosis (e.g. siRNA targeting RNR1). These concerns need therefore to be addressed by authors before resubmitting a revised version of the manuscript as they would strengthen the conclusion. All the other concerns raised by the reviewers should be addressed as well. We, thus, encourage you to submit a revised version of the manuscript back to LSA that responds to all of the reviewers' points.

Thank you for this interesting contribution to Life Science Alliance. We are looking forward to receiving your revised manuscript.

Sincerely,

B. MANUSCRIPT ORGANIZATION AND FORMATTING:

Reviewer #2 (Comments to the Authors (Required)):

In this manuscript, Tarangelo et al. explored how stabilization of p53 and induction of p21 promote ferroptosis resistance/delay ferroptosis in HT-1080 cells. They show that inhibition of RNR-dependent nucleotide metabolism maintains the intracellular GSH levels, reduces lipid peroxidation, and causes a reduction in ferroptosis. The study is well-designed; the experiments are rigorously conducted; and, the results support the conclusions. I provide specific points below that will improve the manuscript and the rigor of the work.

1) The authors infer lipid peroxide levels using C11 BODIPY and make significant conclusions based on these imaging experiments. Prime lipid targets that are prone to peroxidation have been proposed and have been quantified directly. Although, I do acknowledge that these measurements are technically challenging, the incorporation of such direct measurements and investigating the extent of the changes in lipid peroxide levels would be much informative. They would also strengthen the conclusions made using C11 BODIPY.

2) The model presented in Fig. 6 and the relevant hypothesis are tested in HT-1080 cells. Are there other suitable cell lines that could be used to study ferroptosis and test the present model on the mechanistic involvement of p53 activation in ferroptosis? Based on the broad discussions on the applicability of these findings on cancer therapy, I believe testing the generalizability of the results in HT-1080 cells is warranted.

3) How is the level of dAMP measured in Fig. 3E?

4) How will the RNASeq data be made available? Perhaps, I missed this information but I have not seen instructions on how to access the data in the experimental section.

Reviewer #3 (Comments to the Authors (Required)):

In their manuscript, Tarangelo explore the impact of p53 signaling on the sensitivity to ferroptosis. They determine that p53 and the N-terminus of p21 have independent roles in ferroptosis induction. Using hypothesis driven and unbiased transcriptomic analysis, they find that increased nucleotide metabolism downstream is responsible for suppression of ferroptosis upon p53 stabilization. Specifically, their work indicates that GSH consumption by RNR places stress on GSH synthesis and thereby cooperates with GSH depletion to promote ferroptosis. While somewhat incremental, the observations provided here are well-controlled and compelling. The results will be of interest to the ferroptosis community and more broadly to the cancer community, as many anti-cancer agents inhibit replication and may therefore protect cells from oxidative cell death mechanisms. I have two main comments which I think that the authors could readily address prior to publication:

1. One weakness of the manuscript is that the authors do not consider cell-cycle dependent effects and cannot really disentangle inhibition of replication from inhibition of nucleotide synthesis. Could cells in S-phase and actively replicating their DNA be more susceptible to ferroptosis independent of GSH depletion? Could cells outside of S-phase over-produce nucleotides and become ferroptosis sensitive? The authors may wish to discuss these points in greater detail.

2. As the authors discuss, restoration of cell death upon BSO treatment is a crucial control to show that ferroptosis suppression by p53 stabilization and RNR inhibition is GSH dependent. Authors should include this control for experiments in which RNR1 suppression rescues ferroptosis (e.g. siRNA targeting RNR1). These data are also important to help control for cell cycle phase dependent effects which are otherwise not strongly considered.

Minor Points

1. Authors should be a bit careful about their language in the first few sentences of the introduction. Cells, of course, use nucleotides for much more than DNA synthesis. And of course the rate limiting steps for NTP synthesis are not RNR, as this is downstream of NTP production and the rate limiting for dNTP synthesis. Therefore, it would probably be more accurate to replace nucleotide with "deoxyribonucleotide triphosphates" in these first two lines.
2. The statement "Thus, the p21 N-terminus is required to suppress ferroptosis." is overly broad. Certainly cells lacking the p21 N-terminus can undergo ferroptosis (as the authors show some of this is p53 dependent). The authors should revise to make a more accurate statement.
3. Authors should provide analyzed RNAseq data presented in Figure 3 as a supplementary dataset or indicate that these data have been publicly deposited.
4. Authors state "RNR can be directly inhibited using the small molecules gemcitabine (Gem)". This is not widely appreciated to be the predominant mechanism of action of Gem, which is to inhibit elongation during replication. Therefore the effects of Gem may be indirect via polymerase inhibition.

SCOTT J. DIXON
Associate Professor
Department of Biology

We are grateful to the Reviewers for their comments on the manuscript. Below, we address each comment in turn. Please note that changes in the revised main text that address specific Referee comments are highlighted in that document in blue. Additionally, we have made small changes throughout the text to improve the clarity of the work and conform with formatting requirements that are not individually highlighted.

Referee comments and replies:

Reviewer #1:

In this manuscript, Tarangelo et al. explored how stabilization of p53 and induction of p21 promote ferroptosis resistance/delay ferroptosis in HT-1080 cells. They show that inhibition of RNR-dependent nucleotide metabolism maintains the intracellular GSH levels, reduces lipid peroxidation, and causes a reduction in ferroptosis. The study is well-designed; the experiments are rigorously conducted; and, the results support the conclusions. I provide specific points below that will improve the manuscript and the rigor of the work.

Author Response: We thank the Reviewer for the positive comments on the manuscript.

1) The authors infer lipid peroxide levels using C11 BODIPY and make significant conclusions based on these imaging experiments. Prime lipid targets that are prone to peroxidation have been proposed and have been quantified directly. Although, I do acknowledge that these measurements are technically challenging, the incorporation of such direct measurements and investigating the extent of the changes in lipid peroxide levels would be much informative. They would also strengthen the conclusions made using C11 BODIPY.

Author Response: We thank the Reviewer for this suggestion. As the Reviewer notes, these experiments are incredibly challenging. We made extensive efforts to quantify lipid abundance using an LC-MS approach with our co-authors in the Long lab. Our strategy was to search for the loss of specific lipid species upon erastin2 treatment, under the assumption that these would be destroyed during the process of ferroptosis. We were unsuccessful. Using LC-MS, authentic standards, and MS/MS analysis, we could reliably detect an abundant polyunsaturated fatty acid (PUFA)-containing phosphatidylethanolamine phospholipid species we had reason to believe from published studies (e.g., Kagan et al., 2017, *Nat Chem Biol*) was likely be oxidatively destroyed during ferroptosis: PE(C18:0/C20:4). To our surprise, erastin2 treatment did not reduce the abundance of this species, and genetic silencing of *RRM1* resulted in a basal increase in the abundance of this species (see below, each individual datapoint represents a separate biological replicate).

Our preliminary attempts to more fully decipher these observations suggests that there may be interesting and important differences in lipid metabolism depending on whether ferroptosis is induced by direct GPX4 inhibition (as in previous published papers) versus cystine deprivation. These conditions appear to have diverging effects on lipid abundances. This is the focus of ongoing work and, we would like to respectfully suggest, beyond our ability to elucidate here in the present manuscript, whose focus in any case is not on lipid metabolism *per se*.

That said, we now more clearly note in the revised manuscript the tentative nature of our conclusions concerning the accumulation of lipid peroxidation. We hope this is sufficient to satisfy the Reviewer.

2) The model presented in Fig. 6 and the relevant hypothesis are tested in HT-1080 cells. Are there other suitable cell lines that could be used to study ferroptosis and test the present model on the mechanistic involvement of p53 activation in ferroptosis? Based on the broad discussions on the applicability of these findings on cancer therapy, I believe testing the generalizability of the results in HT-1080 cells is warranted.

Author Response: This is an excellent suggestion. In the revised manuscript we now extend our analysis to include results obtained using p53 wild-type Caki-1 cells and p53 null H1299 cell lines. Consistent with our expectations, gemcitabine pretreatment inhibited erastin-2-induced ferroptosis in both cell lines, while nutlin-3 was only effective at attenuating ferroptosis in Caki-1 cells. We have an additional layer of controls built into these experiments, with every condition tested with or without the ferroptosis-specific inhibitor ferrostatin-1 (Fer-1). This condition provides a baseline for non-ferroptotic cell death caused by the nutlin-3 or gemcitabine pretreatments. These data are presented in new Fig. S3:

3) How is the level of dAMP measured in Fig. 3E?

Author Response: We thank the Reviewer for catching this oversight. Unaccountably, we forgot to include these methods in the original submission. This is corrected in the revised manuscript under the heading “Steady state metabolite analysis”.

4) How will the RNASeq data be made available? Perhaps, I missed this information but I have not seen instructions on how to access the data in the experimental section.

Author Response: The Reviewer was not mistaken; we had initially been uncertain at what stage to make our data publicly available. Processed read counts for all experimental conditions and biologically replicates reported in the manuscript are now freely accessible online at Mendeley Data (<http://dx.doi.org/10.17632/gbvp7h5h8j.1>).

Please note here that it was originally our intention to submit our RNA sequencing data to the gene expression omnibus (GEO). However, we realized during this re-submission process that we erroneously did not retain on file the raw read

count (.bam) files necessary for this submission. We have discussed this issue with the Editor and received approval for the above approach to making our processed read counts available online.

Reviewer #2:

In their manuscript, Tarangelo explore the impact of p53 signaling on the sensitivity to ferroptosis. They determine that p53 and the N-terminus of p21 have independent roles in ferroptosis induction. Using hypothesis driven and unbiased transcriptomic analysis, they find that increased nucleotide metabolism downstream is responsible for suppression of ferroptosis upon p53 stabilization. Specifically, their work indicates that GSH consumption by RNR places stress on GSH synthesis and thereby cooperates with GSH depletion to promote ferroptosis. While somewhat incremental, the observations provided here are well-controlled and compelling. The results will be of interest to the ferroptosis community and more broadly to the cancer community, as many anti-cancer agents inhibit replication and may therefore protect cells from oxidative cell death mechanisms. I have two main comments which I think that the authors could readily address prior to publication:

1. One weakness of the manuscript is that the authors do not consider cell-cycle dependent effects and cannot really disentangle inhibition of replication from inhibition of nucleotide synthesis. Could cells in S-phase and actively replicating their DNA be more susceptible to ferroptosis independent of GSH depletion? Could cells outside of S-phase over-produce nucleotides and become ferroptosis sensitive? The authors may wish to discuss these points in greater detail.

Author Response: The Reviewer raises excellent points. It is difficult to disentangle these processes. In results reported in Fig. 5F of the original and revised submission, we do overexpress RRM1/2 sub-units in arrested cells and find that this is not sufficient to promote ferroptosis. However, again, this is a difficult experiment to disentangle because RNR itself may lack the requisite upstream precursors necessary for its own activity in arrested cells where the upstream genes are themselves downregulated.

In the revised Discussion we state our proposed model more tentatively, and now clearly discuss the possible caveats to our model suggested by the Reviewer as follows:

“RNR-dependent nucleotide synthesis may therefore compete with GPX4-dependent lipid hydroperoxide reduction for use of the same co-substrate, namely GSH. We find that blockade of de novo GSH synthesis eliminates the protective effects of genetic or pharmacological RNR inhibition on ferroptosis. Based on these results, we propose that inhibition of RNR (or upstream enzymes) leads to accumulation of GSH which can be re-purposed for use by GPX4 to prolong the inhibition of ferroptosis under conditions where new glutathione synthesis is otherwise impossible (Figure 6). An important caveat of this model is that it may be difficult to disentangle the effects of direct RNR inhibition from more global cell cycle arrest. For example, p53 stabilization results in downregulation of numerous metabolic genes, and we observe that overexpression of RNR sub-units alone is not sufficient to re-sensitize arrested cells to ferroptosis, perhaps due to the absence of upstream precursors that would allow RNR to operate at full capacity. Regardless, the possibility that downregulation of other redox-dependent genes or process upon p53 stabilization, or in response to other conditions leading to cell cycle arrest, can promote GSH accumulation and inhibition of ferroptosis cannot be excluded.”

2. As the authors discuss, restoration of cell death upon BSO treatment is a crucial control to show that ferroptosis suppression by p53 stabilization and RNR inhibition is GSH dependent. Authors should include this control for experiments in which RNR1 suppression rescues ferroptosis (e.g. siRNA targeting RNR1). These data are also important to help control for cell cycle phase dependent effects which are otherwise not strongly considered.

Author Response: As suggested by the Reviewer, we now show in new Fig. S4B that BSO can revert the protective effect of si-*RRM1* treatment, as predicted from results obtained with small molecule inhibitors.

Concerning these results, we now write in the revised results that: “The ability of *RRM1* silencing to inhibit erastin2-induced ferroptosis was reverted by co-treatment with BSO, consistent with a glutathione-dependent mechanism (**Figure S4B**)”. Note here that for this experiment we employed a higher erastin2 concentration (2 μ M) than in previous experiments, such that the protection afforded by *RRM1* silencing was somewhat reduced. Not shown in the revised manuscript, we did confirm that our si*RRM1* treatments continue to effectively reduce expression of this gene.

Minor Points

1. Authors should be a bit careful about their language in the first few sentences of the introduction. Cells, of course, use nucleotides for much more than DNA synthesis. And of course the rate limiting steps for NTP synthesis are not RNR, as this is downstream of NTP production and the rate limiting for dNTP synthesis. Therefore, it would probably be more accurate to replace nucleotide with "deoxyribonucleotide triphosphates" in these first two lines.

Author Response: We thank the Reviewer for this excellent suggestion and have revised the Introduction accordingly. We have removed the statement about the rate limiting step. We have also expanded the first line to note some of the myriad other roles of nucleotides in the cell. The first lines of the revised Introduction now read as follows:

“Nucleotides are required for various metabolic processes in the cell, including energy metabolism, phospholipid synthesis, N-glycosylation, and of course the synthesis of DNA and RNA (Lane & Fan, 2015). De novo nucleotide synthesis in mammalian cells is accomplished through a multi-step pathway that converts ribonucleotide triphosphates (NTPs) to deoxyribonucleotide triphosphates (dNTPs) (Elledge et al, 1992; Tran et al, 2019). dNTPs are essential for DNA replication.”

2. The statement "Thus, the p21 N-terminus is required to suppress ferroptosis." is overly broad. Certainly cells lacking the p21 N-terminus can undergo ferroptosis (as the authors show some of this is p53 dependent). The authors should revise to make a more accurate statement.

Author Response: We completely agree and are grateful to the Reviewer for noting this error. The key words "...for p21..." were missing from that sentence, which in the revised results now reads more correctly as: “Thus, the p21 N-terminus is required *for p21* to suppress ferroptosis.” (*emphasis added here*)

3. Authors should provide analyzed RNAseq data presented in Figure 3 as a supplementary dataset or indicate that these data have been publicly deposited.

Author Response: We agree. All RNA sequencing data for all experimental conditions reported in the manuscript is now accessible online at Mendeley Data (<http://dx.doi.org/10.17632/gbvp7h5h8j.1>).

As mentioned above, please note here that it was originally our intention to submit our RNA sequencing data to the gene expression omnibus (GEO). However, we realized during this re-submission process that we erroneously did not retain the necessary raw read count (.bam) files necessary for this submission. We have discussed this issue with the Editor and received approval for the above approach to making our processed read counts available online.

4. Authors state "RNR can be directly inhibited using the small molecules gemcitabine (Gem)". This is not widely appreciated to be the predominant mechanism of action of Gem, which is to inhibit elongation during replication. Therefore the effects of Gem may be indirect via polymerase inhibition.

Author Response: This is a good point. We would, however, like to note that we do see the same effect as Gem with a different proposed RNR inhibitor, hydroxyurea. Nonetheless, to account for this possible alternative hypothesis mentioned by the Reviewer we now write in the revised Discussion as follows:

“By contrast, the effects of anti-cancer agents that seek to induce ferroptosis via GSH depletion (e.g. cyst(e)inase, (Cramer et al, 2017)) could be blunted if combined with chemotherapeutics like gemcitabine and hydroxyurea, which inhibit the consumption of GSH by nucleotide metabolic processes and also slow DNA replication by blocking elongation (Heinemann et al., 1990; Plunkett et al, 1995).”

December 13, 2021

RE: Life Science Alliance Manuscript #LSA-2021-01157R

Dr. Scott J Dixon
Stanford University
Department of Biology
327 Campus Dr., Room 104
Bass Biology Building
Stanford, CA 94305

Dear Dr. Dixon,

Thank you for submitting your revised manuscript entitled "Nucleotide Biosynthesis Links Glutathione Metabolism to Ferroptosis Sensitivity". We would be happy to publish your paper in Life Science Alliance pending final revisions necessary to meet our formatting guidelines.

- please upload your main and supplementary figures as single files;
- please add the Twitter handle of your host institute/organization as well as your own or/and one of the authors in our system
- please make sure the author order in your manuscript and our system match
- please consult our manuscript preparation guidelines <https://www.life-science-alliance.org/manuscript-prep> and make sure your manuscript sections are in the correct order and labeled correctly
- please be sure that all authors are listed in the Authors contribution section in the manuscript text
- we encourage you to revise the figure legend for Figure 4 such that the figure panels are introduced in an alphabetical order
- please add your main and supplementary figure legends to the main manuscript text after the references section;
- Please indicate molecular weight next to each protein blot
- please provide Data Availability as a separate section

FIGURE CHECKS:

- scale bars for figure 5C are missing, please provide them.

A. FINAL FILES:

B. MANUSCRIPT ORGANIZATION AND FORMATTING:

Sincerely,

Reviewer #2 (Comments to the Authors (Required)):

The authors have addressed my suggestions and comments; thank you.

Reviewer #3 (Comments to the Authors (Required)):

The authors have addressed all of my comments and this is an excellent manuscript for publication.

January 11, 2022

RE: Life Science Alliance Manuscript #LSA-2021-01157RR

Dr. Scott J Dixon
Stanford University
Department of Biology
327 Campus Dr., Room 104
Bass Biology Building
Stanford, CA 94305

Dear Dr. Dixon,

Thank you for submitting your Research Article entitled "Nucleotide Biosynthesis Links Glutathione Metabolism to Ferroptosis Sensitivity". It is a pleasure to let you know that your manuscript is now accepted for publication in Life Science Alliance. Congratulations on this interesting work.

DISTRIBUTION OF MATERIALS:

Again, congratulations on a very nice paper. I hope you found the review process to be constructive and are pleased with how the manuscript was handled editorially. We look forward to future exciting submissions from your lab.

Sincerely,
